Discovery of diverse Pectocaris species at the Cambrian series 2 Hongjingshao formation Xiazhuang section (Kunming, SW China) and its ecological, taphonomic, and biostratigraphic implications

Jin Changfei 1 2
Chen Hong 3
Mai Huijuan 1 2
Hou Xianguang 1 2 xghou@ynu.edu.cn
Yang Xianfeng 1 2
Zhai Dayou 1 2 dyzhai@ynu.edu.cn
1 Yunnan Key Laboratory for Palaeobiology, Institute of Palaeontology, Yunnan University , Kunming , China
2 MEC International Joint Laboratory for Palaeobiology and Palaeoenvironment, Yunnan University , Kunming , China
3 School of Biological Sciences and Technology, Liupanshui Normal University , Liupanshui , China
Piñeiro Graciela
Electronic publication date: 2024 Apr 15
Publication date: 2024
Volume: 12
Electronic Location ID: e17230
Received 2023 Nov 29; Accepted 2024 Mar 21
Copyright: © 2024 Jin et al.
Copyright year: 2024
Copyright holder: Jin et al.
License: This is an open access article distributed under the terms of the Creative Commons Attribution License, which permits unrestricted use, distribution, reproduction and adaptation in any medium and for any purpose provided that it is properly attributed. For attribution, the original author(s), title, publication source (PeerJ) and either DOI or URL of the article must be cited.
License URL: https://creativecommons.org/licenses/by/4.0/

Keywords: Arthropod, Niche differentiation, Pectocaris paraspatiosa, Hongjingshao formation, Xiazhuang assemblage

Funding: Yunnan Provincial Grants Nos. 202101AT070158, 202305AB350006, 202301AS070049, YNWR-QNBJ-2019-295 Yunnan Key Laboratory for Palaeobiology, Yunnan University Liupanshui Normal University No. LPSSYKYJJ202210 Yunnan Science & Technology Champion Project 202305AB350006 This work was supported by the Yunnan Provincial Grants (Nos. 202101AT070158, 202305AB350006, 202301AS070049, YNWR-QNBJ-2019-295), the “Open for collaboration” grant from Yunnan Key Laboratory for Palaeobiology, Yunnan University and High-Level Talents of Liupanshui Normal University (No. LPSSYKYJJ202210), and Yunnan Science & Technology Champion Project 202305AB350006. The funders had no role in study design, data collection and analysis, decision to publish, or preparation of the manuscript.

==============================
Pectocaris species are intermediate- to large-sized Cambrian bivalved arthropods. Previous studies have documented Pectocaris exclusively from the Cambrian Series 2 Stage 3 Chengjiang biota in Yu’anshan Formation, Chiungchussu Stage in SW China. In this study, we report Pectocaris paraspatiosa sp. nov., and three other previously known Pectocaris from the Xiazhuang section in Kunming, which belongs to the Hongjingshao Formation and is a later phase within Cambrian Stage 3 than the Yu’anshan Formation. The new species can be distinguished from its congeners by the sparsely arranged endopodal endites and the morphologies of the abdomen, telson, and telson processes. We interpret P. paraspatiosa sp. nov. as a filter-feeder and a powerful swimmer adapted to shallow, agitated environments. Comparison among the Pectocaris species reinforces previous views that niche differentiation had been established among the congeneric species based on morphological differentiation. Our study shows the comprehensive occurrences of Pectocaris species outside the Chengjiang biota for the first time. With a review of the shared fossil taxa of Chengjiang and Xiaoshiba biotas, we identify a strong biological connection between the Yu’anshan and Hongjingshao Formations.

Introduction

Pectocaris species are intermediate- to large-sized bivalved arthropods in the Cambrian ocean (Hou, 1999; Hou, Bergström & Xu, 2004; Jin et al., 2021). The genus currently comprises three species, P. spatiosa Hou, 1999, P. eurypetala (Hou & Sun, 1988), and P. inopinata Jin et al., 2021. All of them were so far only known from the Cambrian Series 2 Stage 3 Chengjiang Lagerstätte in eastern Yunnan, Southwest China. They were characterized by the large carapace covering about half of the body length, a great number of short stout body segments, densely arranged comb-like appendages with multiple podomeres and broad telson processes (Hou, Bergström & Xu, 2004). Pectocaris had been considered as branchiopod crustaceans and resolved within Hymenocarina later, thus showing affinity with the Mandibulata (Hou, 1999; Hou, Bergström & Xu, 2004; Aria & Caron, 2017; Izquierdo-López & Caron, 2022a). However, the key apomorphies of Mandibulata, for instance, a third appendage specialized as the mandible, has not been found in Pectocaris. Therefore, their phylogenetic position within the arthropod tree remains uncertain. In terms of ecology, Pectocaris have been considered to be a swimming species in view of the well-developed abdomen and telson processes (Hou & Sun, 1988; Hou, 1999; Hou, Bergström & Xu, 2004; Jin et al., 2021). Although P. inopinata was interpreted as a predator and/or scavenger (Jin et al., 2021), the other two species of the genus have been regarded as filter feeders (Hou, Bergström & Xu, 2004).

The geographical distribution and ecological niche of Pectocaris remain poorly known. A number of non-trilobite arthropods initially found in the Chengjiang Lagerstätte, including Isoxys, Leanchoilia, Misszhouia, Naraoia, and Xandarella, have recently been found in contemporary Lagerstätten in South China including the Qingjiang (Fu et al., 2019) and Fandian (Du et al., 2020) Lagerstätten. Some bivalved arthropods, such as Isoxys, are found across the Cambrian period, spanning tens of millions of years (Williams, Siveter & Peel, 1996; Hu et al., 2007; García-Bellido et al., 2009; García-Bellido, Vannier & Collins, 2009; Sun et al., 2022; Ma et al., 2023). In addition to retrieving bio-stratigraphical connections among Lagerstätten, these findings would also indicate that the above mentioned arthropods were ecologically tolerant, or had strong dispersal ability, or that the related lineage (in the case of Isoxys) successfully evolved through time and adapted to the changing environment. But for Pectocaris, none of its species has previously been found outside the Chengjiang Lagerstätte, and the genus seems to be temporal‒spatially restricted.

Recent research in the Xiaoshiba Lagerstätte shows that it has some arthropod genera in common with the Chengjiang Lagerstätte, such as Fuxianhuia and Chengjiangocaris, and has been considered as the temporal continuation of the Chengjiang community (Yang et al., 2013, 2016a; Lan et al., 2018; Yang et al., 2018). Zeng et al. (2014) reported several specimens that resembled Pectocaris and/or Jugatacaris, the latter being phylogenetically close to Pectocaris according to Jin et al. (2021). Therefore, it is reasonable to suspect that the Pectocaris species and their close relatives might have continued surviving after the Chengjiang community. Investigating the distribution of Pectocaris among different Cambrian Lagerstätten would reveal its temporal and spatial distributions, and would improve the understanding of the evolution and environmental adaptation of this group and other Cambrian bivalved arthropods with similar morphologies.

In this article, we report Pectocaris paraspatiosa sp. nov. along with several other previously known species of the genus, based on newly collected material from the Xiazhuang section (Fig. 1). Based on the presence of the Pectocaris species in the Xiazhuang assemblage and the morphological difference among Pectocaris species, we discuss the paleoecological, taphonomic, and biostratigraphic significances of this taxon.

Figure 1 Distribution of the main localities of the Chengjiang biota, Yunnan Province and the position of the specimen in the stratum.

(A) Black stars indicate fossil sites yielding Pectocaris in previous studies (Hou & Sun, 1988; Hou, 1999; Hou, Bergström & Xu, 2004; Jin et al., 2021). Red star indicates the fossil locality investigated in the present study (Xiazhuang section) (redrawn in Hou et al., 2017). (B) Stratigraphic column of the Xiazhuang section (redrawn in Zeng et al., 2014). Red arrowhead indicates the position of the specimen analyzed in the present study.

Materials and Methods

Specimens analyzed herein were collected from the Xiazhuang section (cf. Zeng et al., 2014) that lies to the northwest of the Guanshan reservoir, Xiazhuang village, Chenggong district, Kunming, China (Fig. 1). This section includes the Yu’anshan Formation at its lower part and the Hongjingshao Formation at its upper part, both belonging to Cambrian Series 2 (Zeng et al., 2014). In the Hongjingshao Formation, we obtained abundant specimens of Hongshiyanaspis, Fuxianhuia, Kutorgina and Paraselkirkia (cf. Zeng et al., 2014), which are characteristic elements of this formation. In addition, 36 specimens of bivalved arthropods are found to co-occur with the above taxa and are reported in this study.

In the laboratory, fossil specimens were excavated with steel needles, and were photographed with a Leica M205C fluorescence-microscope and a Canon camera equipped with a 100 mm macro lens. Line drawings of the specimens were made with the aid of a camera lucida attached to a NIKON SMZ 1270 stereomicroscope. All the specimens analyzed in this article have been housed at the Yunnan Key Laboratory for Palaeobiology, Institute of Palaeontology, Yunnan University (YKLP).

The electronic version of this article in Portable Document Format (PDF) will represent a published work according to the International Commission on Zoological Nomenclature (ICZN), and hence the new names contained in the electronic version are effectively published under that Code from the electronic edition alone. This published work and the nomenclatural acts it contains have been registered in ZooBank, the online registration system for the ICZN. The ZooBank LSIDs (Life Science Identifiers) can be resolved and the associated information viewed through any standard web browser by appending the LSID to the prefix http://zoobank.org/. The LSID for this publication is: LSID: urn:lsid:zoobank.org:pub:08180F90-0FD6-4191-9059-38C529AB8EEA. The online version of this work is archived and available from the following digital repositories: PeerJ, PubMed Central SCIE and CLOCKSS.

Systematic paleontology

Phylum ARTHROPODA von Siebold, 1848

Class UNCERTAIN

Order PECTOCARIDIDA Simonetta and Delle Cave, 1975

Family PECTOCARIDIDAE Hou, 1999

Genus PECTOCARIS Hou, 1999

Type species. Pectocaris spatiosa Hou, 1999 (see Hou, 1999, Figure 6).

Other species. Pectocaris eurypetala (Hou & Sun, 1988), Pectocaris inopinata Jin et al., 2021, Pectocaris paraspatiosa sp. nov.

Diagnosis (amended after Jin et al., 2021). Intermediate- to large-sized Cambrian bivalved arthropods. Carapace outline sub-parallelogram or sub-elliptical, devoid of ornaments and marginal spines, covering slightly more than half of body length. Stalked eyes and anterior end of head usually protruding beyond carapace. Trunk usually with over 40 segments each much wider than long. Trunk appendages densely arranged, with flap-like exopods carrying short setae and multi-segmented endopods bearing setulose endites. Telson elongate, connected to a pair of broad lateral processes via small sub-triangular sclerites.

Pectocaris paraspatiosa sp. nov.

Figs. 2‒3

Figure 2 Pectocaris paraspatiosa sp. nov. (YKLP 16289a and b, holotype) from the Hongjingshao Formation at Xiazhuang section.

(A) Part, overview. Note that the carapace is invert by 180° relative to the trunk, with the anterior end on the right and the dorsal margin at the bottom. Arrow indicates anterior direction of the carapace. (B) Counterpart, overview. (C) Details of two endopods (rectangle in A). Red arrowheads indicate setae on two of the endite. (D) Exopod setae (rectangle in A). (E) Trunk appendages (rectangle in B). (F) Details of one of the endopods (rectangle in B), showing seven endites (arrowed; with two endites enlarged in the small inset). (G) Details of two setulose endites (small rectangle in right-upper part of B). (H) Exopod setae (rectangle in E). (I) Setulose endites of the endopod (rectangle in E). Abbreviations: am, anterior margin of carapace; dm, dorsal margin of carapace; en, endopod; lv, left valve; rs, rod-like structure; rv, right valve; sn, the nth body segment (counted from posterior to anterior); st, seta(e); tpm, posterior margin of telson; ts, telson.

Figure 3 Pectocaris paraspatiosa sp. nov. from the Hongjingshao Formation at Xiazhuang section.

(A and B) Holotype (YKLP 16289a and b), explanatory drawings of the overviews in Figs. 2A and 2B, respectively. (C) Explanatory drawing of appendage enlarged in A (rectangle in A). (D) Explanatory drawing of appendage enlarged in B (rectangle in B). (E and F) YKLP 16290 (paratype), light photo and explanatory drawing. A trilobite (Hongshiyanaspis) is preserved with this specimen (adjacent to its telson processes). (G and H) YKLP 16291 (paratype), light photo and explanatory drawing. (I and J) YKLP 16292 (paratype), light photo and explanatory drawing. (K and L) YKLP 16293 (paratype), light photo and explanatory drawing. Abbreviation: sts, sub-triangular sclerite at the basal part of the telson processes.

Etymology: Prefixion “Para-” means the new species is similar to Pectocaris spatiosa Hou, 1999 in appearance.

Type specimens. Holotype: YKLP 16289 (Figs. 2, 3A and 3B); Paratypes: YKLP 16290, YKLP 16291, YKLP 16292, YKLP 16293 (Figs. 3E, 3G, 3I and 3K).

Other material examined. YKLP 16294, YKLP 16295, YKLP 16296, YKLP 16297, YKLP 16298, YKLP 16299, YKLP 16300 (see Supplemental Material).

Diagnosis. Intermediate-sized Pectocaris species. Carapace outline sub-parallelogram. Trunk segments sub-equal in width but progressively longer posteriorly and without sudden narrowing towards telson. Telson processes separated and each almost straight on both outsides. Endites slender, more sparsely spaced than congeners, with gap between endites sub-equal to or slightly larger than width of endite.

Preservation. All the 12 specimens studied herein are incomplete. The holotype (Figs. 2A, 2B, 3A and 3B) preserves the overall outline of carapace, about 19 trunk segments, telson, and some appendages. The carapace is inverted by 180° relative to trunk, presumably due to taphonomic process. The appendages are somewhat dislocated yet with fine details. The other 11 ones preserve the posterior part of the trunk and in some specimens also the telson processes are present. Regarding its preservation, the holotype has the carapace laterally compressed, while the trunk is dorso-ventrally compressed, judging from the shape and width of the telson and the lateral position of the notches which would hold the telson processes if the specimen was complete. Among the other specimens, which only have trunk segments preserved, 10 are preserved in dorsal‒ventral aspect (Figs. 3E, 3G, 3I, 3K and Supplemental Material), and one is preserved in oblique-lateral aspect judged from the widths of the telson and the telson processes (see Supplemental Material, YKLP 16299).

Remarks. We interpret the carapace of the holotype as being inverted by 180° for several reasons. Firstly, in general, the most anterior of the carapace in the species of Pectocaris is usually narrower and the posterior end is broader (cf. the left and the right end of carapace in Figs. 2A and 3A) (Hou, 1999; Hou, Bergström & Xu, 2004; Fu & Zhang, 2011; Jin et al., 2021). Secondly, the dorsal margin of the carapace of Pectocaris is often nearly straight while the ventral margin is usually convex. Thirdly, the gently curved trunk of the holotype, which overlain the anterior margin of the carapace, can be better explained as a relative rotation between the trunk and the carapace (as seen in other species of Pectocaris), rather than a dorsal-ward dislocation of the trunk.

Description. The carapace is elongate (Figs. 2A and 2B), with comparatively straight dorsal and ventral margins. The posterior margin has shaped postero-dorsal part showing appearance of being “cut off”, leaving a short straight edge (Fig. 2A). The postero-dorsal angle is about 150°. A straight rod-like structure in the posterior part of the carapace that connects the dorsal and ventral margins is interpreted as the breakage of the carapace. Marginal spines and ornaments are absent.

About 15 posterior trunk segments are observed in YKLP 16289a (part of holotype) while four additional anterior segments were revealed in YKLP 16289b (counterpart of holotype) by manual preparation (Figs. 2A and 2B). Fewer trunk segments were preserved in other specimens. Each segment is sub-rectangular or trapezoidal in shape, with no dorsal or lateral spines (Figs. 2A, 2B, 3A and 3B). Segments are sub-equal in width and gradually narrow towards the telson.

The telson is sub-trapezoidal, with anterior part sub-equally as wide as the last abdominal segments, while its posterior part is significantly wider, interrupted by small postero-lateral notches (Figs. 2B, 3B and 3E‒3L). A small sub-triangular sclerite that possibly denotes acute lateral extensions (Izquierdo-López & Caron, 2022b) is present in each notch, connecting the telson with broad, sub-divided telson processes (Figs. 3E‒3L). A pair of shallow grooves are present on the telson processes.

Trunk appendages are only preserved in the holotype (Figs. 2 and 3A‒3D), all being incomplete and more or less dislocated from in vivo positions, the longest one consisting of more than 24 podomeres (Fig. 2E). Each podomere bears one sub-quadrate or slightly trapezoidal endite carrying up to six slender apical setae (Figs. 2C, 2F, 2G and 2I). Endite setae in the proximal part of the limb (Figs. 2G and 2I) are somewhat thicker than those in the distal part (Figs. 2C and 2F). The distance between endites is sub-equal to, or slightly larger than the width of each endite (Figs. 2C and 2F). The overall shape of the exopod is not resolved, but it can be seen to bear densely arranged, short marginal setae (Figs. 2D and 2H).

Differential diagnosis. The present species is similar to its congeners and Jugatacaris agilis Fu & Zhang, 2011 in the outline of carapace (Hou, 1999; Hou, Bergström & Xu, 2004; Fu & Zhang, 2011; Jin et al., 2021), such as the narrowly rounded anterior end and the “cut-off” shaped postero-dorsal edge. Like other Pectocaris species, P. paraspatiosa sp. nov. lacks the dorsal fin-like structure which is characteristic of J. agilis (Fu & Zhang, 2011). The Waptia-like bivalved arthropod Xiazhuangocaris chenggongensis from the same section (Zeng et al., 2020) can be readily recognized by the prominent anterior notch and narrow anterior tip of the carapace.

The abdomen of P. paraspatiosa sp., nov. tends to be sub-equally wide throughout its length which is different from all other Pectocaris species that have posteriorly tapering abdomens.

The endopods of most Pectocaris species and J. agilis all have multiple setae-bearing endites (in P. inopinata, the setulose endites are present in posterior trunk appendages and in proximal section of anterior trunk appendages). However, the endites of the other species compared here are more densely arranged, especially in P. inopinata and J. agilis. Moreover, the endites of P. inopinata are stouter and those of P. eurypetala are sub-trapezoidal and seem to be smaller.

Locality and horizon. Xiazhuang section of Chenggong, Kunming, China. Hongjingshao Formation, Cambrian Stage 3, Series 2.

Other Pectocaris species from the Xiazhuang section

In addition to the 12 specimens of P. paraspatiosa sp. nov., 22 specimens collected from the same stratum were also recognized as Pectocaris species, namely P. eurypetala (two specimens), P. spatiosa (19 specimens), P. inopinata (one specimen). Also, there is one specimen most likely belonging to J. agilis (Fig. 4I). All these were preserved with the posterior part of the trunk and sometimes also with the telson processes, while the anterior parts of their bodies were missing. Among the various morphological features described in previous studies (Hou & Sun, 1988; Hou, 1999; Hou, Bergström & Xu, 2004; Fu & Zhang, 2011, Jin et al., 2021), the most useful traits characterizing the present species include the morphology of the limbs, the abdomen, the telson and the telson processes. The comparison of P. paraspatiosa sp. nov. to its congeners from Chengjiang biota show that these species differ from each other in size, carapace shape, number and length of body segments and the presence of dorsal and lateral spines on the abdominal segments, shape of the telson and telson processes, number of trunk-appendage podomeres, and details of enditic armatures (Table 1; Fig. 5). Based on the materials collected in Xiazhuang section, we distinguish P. eurypetala from other species by the more widely spaced abdominal segments while distinct slender telson and the longer broad telson processes (Figs. 4G, 4H and 5B). Even though the limb endites of P. spatiosa are poorly preserved, it can still be recognized by the broad and short abdomen. Along with the sub-rectangular telson and the broad, paddle like telson processes decorated with longer shallow grooves, there are typical characters for this species (Figs. 4A–4F and 5C). Although there is only one specimen of P. inopinata, the characteristic dorsal and lateral spines on the abdominal segments provide unambiguous evidence of its presence within the present samples (Figs. 4K, 4L and 5D). One specimen is identified as Jugatacaris agilis based on its fused telson processes, a characteristic that is absent in Pectocaris (Figs. 4I and 4J).

Figure 4 Other species of the genus Pectocaris from the Hongjingshao Formation at Xiazhuang section. Each specimen is presented with the light photo on the left and the explanatory drawing on the right.

(A‒F), Pectocaris spatiosa Hou, 1999. (A and B) YKLP 16299; (C and D) YKLP 16300; (E and F) YKLP 16301. (G and H), Pectocaris eurypetala Hou & Sun, 1988, YKLP 16302. (I and J), Jugatacaris? sp., YKLP 16303, note the fused left and right telson processes. (K and L) Pectocaris inopinata Jin et al., 2021, YKLP 16304, note the lateral-dorsal abdominal spines characterizing this species. Additional abbreviations: an, anus.

Table 1 Brief comparison of the abdomen and telson morphologies of Pectocaris.

	Species	
Characters	P. paraspatiosa sp. nov.	P. eurypetala	P. spatiosa	P. inopinata	
Carapace	Sub-parallelogram in lateral view	Sub-parallelogram in lateral view/V-shaped openin in dorsal lateral view	Sub-parallelogram in lateral view	Sub-parallelogram in lateral view	
n of endopodal podomeres	>24	c.40	>20	>19	
Endite	Slender, sparsely spaced, with six spines	Slender, tightly spaced, with six spines	Slender, tightly spaced	Stout, tightly spaced, with six spines	
Terminal claw(s) of endopod	Unknown	Small	Unkown	Small, accompanied by prominent paired spines	
Abdomen segments length	Sub-equal, relatively wide	Increasing posteriorly, relatively wide	Sub-equal, relatively narrow	Increasing posteriorly, relatively wide	
Shape of telson	Sub-trapezium, wide, middle	Sub-rectangle, narrow, long	Sub-rectangle, wide, short	Sub-square, narrow, short	
Telson processes	Blade like, broad	Blade like, slender	Blade like, broad	Paddle like, broad	

Figure 5 Schematic drawings of the posterior part of the body and the endites of Pectocaris species.

Based on Hou & Sun (1988), Hou (1999), Hou, Bergström & Xu (2004), Jin et al. (2021), and the present study. Not to scale.

Discussion

Ecology

Like other species of the genus Pectocaris, the strongly built multi-segmented trunk of P. paraspatiosa sp. nov. attached to broad telson processes (Fig. 5) suggests its capability as a good swimmer. The swim propulsion might be stronger than its congeners in view of the longer abdominal segments that could have provided greater torque for its telson processes in beating water. The setulose exopods could as well provide propulsion, even if it is difficult to quantify due to their fragmental preservation in the present specimens. Such assumptions need to be tested with biomechanical models where the function of the muscles attached to the internal surface of the exoskeletons can be analyzed, which could be left to further work.

The multi-segmented endopods of P. paraspatiosa sp. nov. carrying setulose endites (Figs. 2C, 2F, 2G and 2I), similar to P. eurypetala, P. spatiosa, and J. agilis, suggest a filter-feeding behavior as had been interpreted for the latter three species (Hou, Bergström & Xu, 2004; Fu & Zhang, 2011). The endites of P. paraspatiosa sp. nov. are more sparsely arranged compared with other Pectocaris species and J. agilis. This may imply that the new species could filter larger food particles. In addition, the endite setae on the proximal section of the endopods of P. paraspatiosa sp. nov. are thicker than the more distal ones (Fig. 2I), possibly implying that the proximal endites could process harder food particles. In P. inopinata, the distal six podomeres of the endopod equipped with paired strong claws were interpreted to have been used for ploughing through the sediments for food, for grasping preys, and/or for scratching tissues from carcasses (Jin et al., 2021). Such claws are not observed in P. paraspatiosa sp. nov. (Fig. 2E). Therefore, a filter-feeding behavior is a more appropriate explanation than a predatory or scavenging feeding for this species.

The morphological differences among Pectocaris species (Hou, 1999; Hou, Bergström & Xu, 2004; Jin et al., 2021; this study) indicate that this genus was a polymorphic group adapting to various ecological niches. The co-occurrence of the four species of Pectocaris (Figs. 2‒4) as well as Jugatacaris at the Xiazhuang section reinforces the previous conclusion that taxonomically close Cambrian arthropods could develop different body and appendage structures in order to establish niche differentiation and thus exploit resources provided by the competitive marine ecosystem (Jin et al., 2021; Zeng et al., 2020). Pectocaris spp., Xiazhuangocaris chenggongensis, and Clypecaris serrata (Yang et al., 2016b; Zeng et al., 2020; this study) further showcase the morphological disparity of the Hymenocarina (see e.g., Izquierdo-López & Caron, 2022a) in the Hongjingshao, which however is less diversified than the hymenocarines in the Chengjiang and the Burgess Shale faunas (Briggs, Erwin & Collier, 1994; Hou et al., 2017; Izquierdo-López & Caron, 2022a). The only verified bivalved arthropods at the Hongjingshao Formation of Xiazhuang section include the Pectocaris species reported in this study and X. chenggongensis described by Zeng et al. (2020), which are intermediate- to large-sized swimmers. Small bivalved arthropods such as bradoriids, Clypecaris and Ercaicunia have not been reported from this site. Considering the generally coarse lithology compared with the mudstones in the Yu’anshan Formation where the Chengjiang biota is preserved, this may denote stronger hydraulic conditions that were unfavorable for the preservation of smaller swimming arthropods.

Taphonomic implications

A noticeable feature of the Pectocaris specimens from the Xiazhuang section is that except for the holotype (Fig. 2) and one P. spatiosa specimen (Figs. 4A and 4B), all other specimens only preserve the posterior part of the trunk, with anterior part of the body and appendages missing. Although one complete individual of X. chenggongensis and one specimen of Jugatacaris? sp. preserved with carapace and appendicular details were reported from the same stratum (Zeng et al., 2014, 2020), such cases have been rare. By contrast, the intermediate- and large-sized hymenocarines in the Cambrian series 2 strata elsewhere, represented by the Pectocaris species and Jugatacaris agilis from the Chengjiang biota (Yu’anshan Formation, see Hou, Bergström & Xu, 2004; Fu & Zhang, 2011; Jin et al., 2021), and the pectocaridid-like arthropod from the Xiaoshiba biota, usually preserve both the anterior and the posterior parts of the body, although incomplete specimens with only the posterior part of the trunk can also be found (e.g., the Meishucun section, see Hou & Sun, 1988). Considering the combined effects of living environment, burial environment, burial process and individual size on biological preservation (Saleh et al., 2020, 2021, 2022b), our interpretations of taphonomy implications are described below.

The clastic sediments of Hongjingshao Formation at the Xiazhuang section contain a great portion of siltstone and sandstone beds, being generally coarser than the underlying Yu’anshan Formation that is deposited as part of a delta and dominated by mudstones (Zeng et al., 2014; Saleh et al., 2022a; our field observation). This presumably denotes strong hydrodynamics and would result in more intensified mechanical breakage of the thanatocoenosis before burial, as discovered in modern sedimentary systems (e.g., Zhai et al., 2015). Meanwhile, the Hongjingshao stratum lacks the alternating background and event beds, which are characteristic of the Yu’anshan Formation containing soft-bodied Chengjiang biota (Zeng et al., 2014). Previous investigations suggested that post-mortem transport was limited in the event beds, where the thanatocoenosis probably experienced fast, in situ burial (Zhao et al., 2009; Zhao, Zhu & Hu, 2012). We propose that the strong hydraulic disturbance and unfavorable burying mode at the Xiazhuang section were the main causes for the incompleteness of the Pectocaris specimens. The strong hydraulic disturbance, which can be inferred from both the coarse lithology and the incomplete preservation of the Pectocaris specimens, could support our above inference that P. paraspatiosa sp. nov. had been a powerful swimmer adapted to turbulent environments.

Meanwhile, the selective preservation of our Pectocaris specimens may be due to the fact that the anterior part of their body is less resistant to degradation. Observations on relatively complete specimens of P. eurypetala, P. inopinata, and P. spatiosa from the Chengjiang biota revealed clear sclerotization of the posterior part of the trunk protruding beyond the carapace, in contrast to very faint sclerotization in the anterior part of the body protected by the carapace (Hou, Bergström & Xu, 2004; Jin et al., 2021). This is similar to other Cambrian bivalved arthropods, such as Chuandianella ovata Li, 1975 (Zhai et al., 2022), Nereocaris exilis Legg et al., 2012, and Waptia fieldensis Walcott, 1912 (Vannier et al., 2018), probably denoting reaction to the protective carapace for the anterior part and the need for propulsion action for the posterior part of the body, respectively. The trace of the appendages and gut suggest that the carapace of Pectocaris was generally thin and poorly sclerotized/mineralized compared with other bivalved arthropods, such as Chuandianella ovata, Ercaicunia multinodosa Luo & Hu, 1999, and Clypecaris preserved in other Cambrian series 2 strata in eastern Yunnan (Yang et al., 2016b; Hou et al., 2017; Zhai et al., 2019; Liu, Fu & Zhang, 2021). Therefore, we assume that, upon death, the Pectocaris individuals at the Xiazhuang section were probably exposed to certain degree of mechanical and biochemical degradations before being buried, resulting in the absence of the more labile anterior part of body in most of the specimens. Another possible reason is that the molting behavior makes the front part of the body improbable to preserve. However, although interesting to take into account, it is difficult to discuss about this behavior based on the available materials for this paper. Whether the turbulent transportation-accompanying size sorting had occurred for the Pectocaris specimens from the Xiazhuang section is uncertain, because although all the specimens we study are comparatively large late forms, those collected from the finer mudstones in Yu’anshan Formation were dominated by large individuals as well. However, the morphologies of appendage, telson, and telson processes may be influenced by the causes of taphonomy, ontogeny, and sexual dimorphism, which makes it difficult for us to describe and identify new species. The collection of more complete specimens will help to supplement the morphological information and diagnostic characteristic of the species.

Biostratigraphical connection between Hongjingshao and Yu’anshan formations

The Hongjingshao (Member) Formation, which was originally named by Zhang (1966) with type section from the northern hillside between Canglangpu (i.e., Tsanglangpu, now known as Changlongpu) and Hongjingshao (now known as Hongjunshao Village) in Malong County, is characterized by thick sandstone beds intercalated with thin mudstones (Luo, 1994, Yang et al., 2013; Zeng et al., 2014). The underlying Yu’anshan Formation, which was originally named with type section from Mt. Yu’anshan to the west of Kunming City, is characterized by intermediately thick mudstone beds intercalated with sandstone beds (Luo et al., 1982), and in many localities the mudstones consist of frequent alternations of dark-coloured background beds and light-coloured event beds (Zhu, Zhang & Li, 2001; Zhu et al., 2005; Hou et al., 2017). Thereby, as the lower part of the Tsanglangpu Stage and the upper part of the Chiungchussu Stage, respectively, the Hongjingshao and the Yu’anshan Formations had been originally distinguished by lithology. Meanwhile, the Yu’anshan Formation yields the Chengjiang Lagerstätte (Zhang & Hou, 1985; Chen et al., 1996; Hou, Bergström & Xu, 2004; Hou et al., 2017), and the Hongjingshao Formation preserves the Xiaoshiba Lagerstätte (Yang et al., 2013, 2015, 2018) and the Xiazhuang fossil assemblage (Zeng et al., 2014), which is considered contemporary to the lower part of the Xiaoshiba Lagerstätte (Zeng et al., 2017, 2020).

Previous works suggested that both the Xiaoshiba Lagerstätte and the Xiazhuang assemblage are extensions of the Chengjiang Lagerstätte in view of their shared faunal compositions, especially the trilobites (Hou et al., 2017; Zeng et al., 2017). In this study, we make a checklist of shared fossil taxa between the Yu’anshan Formation and the Hongjingshao Formation based on previous references (Table 2), and include the genus Pectocaris in this list in the light of the present study. Such a checklist showcases the biostratigraphical connection between the Hongjingshao Formation (of Tsanglangpu Stage) and the Yu’anshan Formation (of Chiungchussu Stage).

Table 2 Summary of fossil taxa (families and lower ranks) shared by Chengjiang biota, the Yu’anshan Formation (Chiungchussu Stage) and Xiaoshiba biota/Xiazhuang assemblage, the Hongjingshao Formation (Tsanglangpu Stage).

Shared taxon	Species in Yu’anshan Formation	Species in Hongjingshao Formation	
Genus Wiwaxia Walcott, 1911	Wiwaxia papilio Zhang et al., 2015; Zhao et al., 2014	Wiwaxia foliosa Yang et al., 2014	
Genus Kutorgina Billings, 1861	Kutorgina chengjiangensis Zhang et al., 2007	Kutorgina chengjiangensis (Zeng et al., 2014)	
Genus Selkirkia Walcott, 1911	Selkirkia sinica (= Paraselkirkia sinica) (Luo et al., 1999)	Selkirkia sinica Luo & Hu,1999 (Lan et al., 2015)	
Genus Sicyophorus Luo et al., 1999	Sicyophorus rarus Luo et al., 1999	Sicyophorus sp. (Zeng et al., 2014)	
Genus Mafangscolex Hu, 2005	Mafangscolex sinensis Hou & Sun, 1988	Mafangscolex cf. yunnanensis Yang et al., 2020	
Genus Hallucigenia Conway Morris, 1977	Hallucigenia fortis Hou & Bergström, 1995	Hallucigenia? (Zeng et al., 2014)	
Family Luolishaniidae Hou & Bergström, 1995	Luolishania longicruris Hou & Chen, 1989	Collinsium ciliosum Yang et al., 2015	
Genus Pectocaris Hou et al., 1999	P. eurypetala (Hou & Sun, 1988), P. inopinata Jin et al., 2021, P. spatiosa Hou, 1999	P. eurypetala, P. paraspatiosa sp. nov., P. inopinata, P. spatiosa (this study)	
Genus Jugatacaris Fu & Zhang, 2011	Jugatacaris agilis Fu & Zhang, 2011	Jugatacaris? sp. (Zeng et al., 2014)	
Genus Combinivalvula Hou, 1987	Combinivalvula chengjiangensis Hou, 1987	Combinivalvula sp. (Zeng et al., 2014)	
Genus Chengjiangocarididae Hou & Bergström, 1997	Chengjiangocaris longiformis Hou & Bergström, 1991	Chengjiangocaris kunmingensis Yang et al., 2013 Alacaris mirabilis Yang et al., 2018	
Genus Fuxianhuia Hou, 1987	Fuxianhuia protensa Hou, 1987 (Hou & Bergström, 1997)	Fuxianhuia xiaoshibaensis Yang et al., 2013	
Genus Kuanyangia Hupé, 1953	Kuanyangia pustulosa (Lu, 1941), Kuanyangia sp. of Hou & Bergström, 1997 (Hou, Bergström & Xu, 2004; Hou et al., 2017)	Kuanyangia (Sapushania) granulosa Zhang, 1966; Zeng et al., 2014	
Genus Yunnanocephalus Kobayashi, 1936	Yunnanocephalus yunnanensis (Mansuy, 1912; Hou, Bergström & Xu, 2004)	Yunnanocephalus yunnanensis (Mansuy, 1912; Zeng et al., 2014	
Genus Dolerolenus Leanza, 1949	Dolerolenus (Malungia) laevigata Lu, 1961 Dolerolenus (Malungia) malungensis Lu, 1961	Dolerolenus (Malungia) laevigata Lu, 1961 Dolerolenus (Malungia) malungensis Lu, 1961	
Genus Chengjiangaspis Zhang & Lin, 1980	Chengjiangaspis chengjiangensis Zhang & Lin, 1980	Chengjiangaspis chengjiangensis (Hou et al., 2019)	
Genus Hongshiyanaspis Zhang & Lin, 1980	Hongshiyanaspis yiliangensis Zhang et al., 1980	Hongshiyanaspis yiliangensis (Hou et al., 2019)	
Genus Liangwangshania Chen, 2005	Liangwangshania biloba Chen, 2005	Liangwangshania biloba? Chen, 2005	

We concur with Zeng et al. (2014) that the Hongjingshao Formation is diachronous. As litholostratigraphic units, formations and members are defined by lithology, and would basically reflect the sedimentary settings. However, because depositional process can be diachronous, the lithostratigraphic units are not necessarily isochronic. Meanwhile, because faunal communities could potentially survive in different environments with different sedimentary settings, different members or formations in the strata can have the same fossil taxa if they belonged to the same geological age. Hongjingshao Formation is transitional for Cambrian marine communities, also opening up some new genera and species. The extension of some species from the Yu’anshan Formation into the Hongjingshao Formation indicates that these species were managed to survive after the Chengjiang time period despite changes in marine environment, as marked by the lithology (Table 2).

Conclusion

Based on the available materials and their morphological characteristics, we report Pectocaris paraspatiosa sp. nov., as the fourth species of the genus, from the Cambrian Series 2 Hongjingshao Formation at the Xiazhuang section, Kunming, China. The new species can be distinguished from its congeners by a number of features, including the sub-equal widths of posterior trunk segments and their greater length, the absence of dorsal and lateral body spines, the shape of telson, and the sparsely distributed setulose endites on the endopods. We interpret P. paraspatiosa sp. nov. as a powerful swimmer adapted to relatively strong hydraulic condition in shallow water, based on a combination of evidence including the broad telson processes, the strongly built limbless abdominal segments and telson, as well as the coarse lithology in which it is preserved. A filter-feeding strategy is inferred from the multi-segmented endopods carrying setulose endites, which is similar to its congeners.

Comparison among the Pectocaris species suggests that this genus could have been polymorphic and intra-genus niche differentiation was established by morphological differentiation. Such inference is supported by the co-occurrence of the various Pectocaris species at the Xiazhuang section. The Pectocaris species, along with Xiazhuangocaris chenggongensis previously described from the same section, may imply that the agitating environment therein favored intermediate and large swimming hymenocarines to small bivalved arthropods.

We tentatively suggest that the strong hydraulic disturbance and unfavorable burying mode at the Xiazhuang section were the main causes for the incompleteness of the Pectocaris specimens. The different degrees of sclerotization of the anterior and the posterior parts of the trunk, and the weak sclerotization/mineralization of the carapace of Pectocaris, also played a role in selective degradation.

The discovery of Pectocaris species in the Xiazhuang fossil assemblage adds to the list of shared fossil taxa between the Hongjingshao Formation and the underlying Yu’anshan Formation, reinforcing conclusions made by previous studies that both the Xiaoshiba Lagerstätte and the Xiazhuang assemblage were continuing to the Chengjiang biota.

Supplemental Information

Supplemental Information 1 12 posterior trunk segments and telson of Pectocaris paraspatiosa sp. nov.

Supplemental Information 2 About 19 posterior trunk segments and telson of Pectocaris paraspatiosa sp. nov., showing the sub-equal posterior trunk segments which is gradually narrowing towards the telson.

Supplemental Information 3 The incomplete abdomen and the telson of Pectocaris paraspatiosa sp. nov., showing the blade like telson processes.

Supplemental Information 4 The incomplete abdomen and the telson of Pectocaris paraspatiosa sp. nov., showing the sub-equal posterior trunk segments and blade like, broad telson processes.

Supplemental Information 5 The abdomen and telson of Pectocaris paraspatiosa sp. nov., showing the sub-equal posterior trunk segments.

Supplemental Information 6 The abdomen and telson of Pectocaris paraspatiosa sp. nov., showing the sub-trapezoid telson and sub-equal posterior trunk segments.

Supplemental Information 7 The abdomen and telson of Pectocaris paraspatiosa sp. nov.

Supplemental Information 8 The abdomen and telson of Pectocaris paraspatiosa sp. nov., showing the gradually narrowed posterior trunk segments.

Supplemental Information 9 The incomplete abdomen and the telson of Pectocaris paraspatiosa sp. nov.

Supplemental Information 10 The abdomen and telson of Pectocaris paraspatiosa sp. nov., showing the gradually narrowed abdomen.

Supplemental Information 11 Carapace comparison.

We are grateful to Dr. Farid Saleh (Université de Lyon) and Dr. Alejandro Izquierdo-López (University of Toronto) for reviewing this manuscript and providing constructive comments and suggestions. We thank Prof. Peiyun Cong, Mr. Di Wu, Miss. Mengying Yin, Miss. Ting Zhao, and Dr. Yang Zhao (YKLP, Yunnan University) for attending the field work, Prof. Jie Yang, Mr. Wei Li (YKLP, Yunnan University), and Dr. Ruilin Wen (Institute of Geology and Geophysics, Chinese Academy of Sciences) for helpful discussion. Generous support for field investigation from Mr. Kunhong Chen and his colleagues at the Yuhua Subdistrict Administration Office is much appreciated.

Additional Information and Declarations

Competing Interests

Author Contributions

Data Availability

New Species Registration

The authors declare that they have no competing interests.

Changfei Jin conceived and designed the experiments, performed the experiments, analyzed the data, prepared figures and/or tables, authored or reviewed drafts of the article, and approved the final draft.

Hong Chen conceived and designed the experiments, performed the experiments, analyzed the data, prepared figures and/or tables, authored or reviewed drafts of the article, and approved the final draft.

Huijuan Mai conceived and designed the experiments, authored or reviewed drafts of the article, and approved the final draft.

Xianguang Hou conceived and designed the experiments, authored or reviewed drafts of the article, and approved the final draft.

Xianfeng Yang conceived and designed the experiments, authored or reviewed drafts of the article, and approved the final draft.

Dayou Zhai conceived and designed the experiments, authored or reviewed drafts of the article, and approved the final draft.

The following information was supplied regarding data availability:

The specimens supporting the characteristics of the new species are not presented in the article, but serve as evidence for the identification of the new species.

All supplementary figures, YKLP 16294a,b; YKLP 16295a,b; YKLP 16296a,b; YKLP 16297; YKLP 16298; YKLP 16299; YKLP 16300, are stored in the Institute of Palaeontology, Yunnan University (YKLP).

The following information was supplied regarding the registration of a newly described species:

Publication LSID:

urn:lsid:zoobank.org:pub:08180F90-0FD6-4191-9059-38C529AB8EEA

Pectocaris paraspatiosa LSID:

urn:lsid:zoobank.org:act:B1FD1F8D-2B3D-4475-BFDA-669D9F11DF09.

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
