# Peer review of "Discovery of diverse Pectocaris species at the Cambrian series 2 Hongjingshao formation Xiazhuang section (Kunming, SW China) and its ecological, taphonomic, and biostratigraphic implications"

_PeerJ, doi:10.7717/peerj.17230_

## Round 0.1 · original submission · Minor Revisions

Dear authors,

We have now two review reports for your article entitled “Discovery of diverse Pectocaris species at the Cambrian Series 2 Hongjingshao Member Xiazhuang section (Kunming, SW China) and its ecological, taphonomical, and biostratigraphical implications”, and both reviewers consider that it deserves to be published if it is improved according to their recommendations. Therefore, please, read carefully the reports and consider each of the comments made by the reviewers in order to make your work publishable in PeerJ.

The most attention has to be paid to the language, as following the suggestions from reviewers. You can also see the annotated PDF that I have attached to this decision letter. Other concerns are referred to a) the need to provide complementary descriptions of some characters present in the studied specimens in order to better justify that they represent a new species of Pectocaris and b) the need for a revision of the suggested hypothesis of a 180° rotation of the carapace, and a more convincing explanation in supporting of such a proposal. If you can strengthen the support for the presence of a new species of P. paraspatiosa in agreement to Reviewer 2 requests, then you have to make a Zoobank registration of the new species following what Reviewer 2 kindly described and I am copying here as a refresh previously to the resubmission of your manuscript:
“New species guidelines
The new species generally complies with PeerJ and ICZN guidelines, but requires the following: 1) Zoobank registration, 2) LSID number and 3) the following text as per PeerJ guidelines:
"The electronic version of this article in Portable Document Format (PDF) will represent a published work according to the International Commission on Zoological Nomenclature (ICZN), and hence the new names contained in the electronic version are effectively published under that Code from the electronic edition alone. This published work and the nomenclatural acts it contains have been registered in ZooBank, the online registration system for the ICZN. The ZooBank LSIDs (Life Science Identifiers) can be resolved and the associated information viewed through any standard web browser by appending the LSID to the prefix http://zoobank.org/. The LSID for this publication is: [INSERT HERE]. The online version of this work is archived and available from the following digital repositories: PeerJ, PubMed Central SCIE and CLOCKSS."”
I consider that there is a big problem to the readers referring the stratigraphic and biostratigraphic arrangement of the purposed new species and the other taxa that you used for comparative purposes (see the annotated pdf that I have provided). I recommend basically that you present the stratigraphic data in an additional figure of a schematic lithostratigraphic section, showing the stratigraphic arrangement for the studied localities or formations and their members, also including the position of the new and the other compared species. You should leave clear if the new finding suggests an eventual diversity of bivalved arthropods within the influence area of the Chengjiang Lagerstätte.

I hope that you consider all the reports as interesting to help you to improve your manuscript, and any concern or disagreement that you could have concerning the requests, you can explain them in the rebuttal letter, and they will be properly considered.

With all my best regards,
Graciela Piñeiro

**Language Note:** The review process has identified that the English language must be improved. PeerJ can provide language editing services - please contact us at copyediting@peerj.com for pricing (be sure to provide your manuscript number and title). Alternatively, you should make your own arrangements to improve the language quality and provide details in your response letter. – PeerJ Staff

·

Basic reporting

This is a new study describing newly discovered arthropod material from the Cambrian of China. The discovery is noteworthy and deserves to be published. The introduction is clear and shows context. The figures are relevant, and the fossils are beautiful.

Experimental design

The study fits within the scope of the journal, and the questions are well-defined filling a knowledge gap on the existence of the bivalved arthropod Pectocaris outside of the Chengjiang Biota. The methods used are standard for this sort of investigation and are described in sufficient detail.

Validity of the findings

All necessary information to replicate the work is provided. And the conclusions are well-stated and linked to the results and introduction.

Additional comments

I enjoyed reviewing this paper. However, prior to its publication, I think there are some moderate comments to account for.

- English language: I believe that the paper needs to be checked by a fluent English speaker to improve clarity and overall flow
- There are a lot of papers published in the past years that are dismissed in the current work and deserve to be included. For example, the shallow high-energy environment of the region is particularly investigated by Saleh et al., 2022 in Nature Communications (The Chengjiang Biota inhabited a deltaic environment). Also, the low preservation potential of swimming taxa in comparison to the rest is investigated in Saleh et al., 2022 in PeerJ (Probability-based preservational variations within the early Cambrian Chengjiang biota (China)). Also see Saleh et al., 2020; and 2021 in EPSL for a broader discussion between Lagerstätten (Taphonomic bias in exceptionally preserved biotas) and (A novel tool to untangle the ecology and fossil preservation knot in exceptionally preserved biotas). Essentially all these papers have discussed elements of depositional environment, and/or preservation that are directly relevant to this paper, particularly to its discussion, and deserve to be included.
- Line 35. Replace “Member of” with “Member of the”
- Line 39. Replace “agitating environment” with “agitated environments”
- Line 42. Replace “to verify the occurrence” with “to show occurrences”
- Line 52. Replace “half length” with “half of the length”
- Line 54. Add reference after “tailfans”.
- Line 55. Replace “with Mandibulata” with “with the Mandibulata”
- Line 59. Should be “a swimming species”
- Line 61. Should be “as a predator and/or scavenger”
- Line 62. Should be “filter feeders”
- Line 67. Add Reference of Ma et al., 2023 in Frontiers in Ecology and Evolution (Ontogeny and brooding strategy of the early Cambrian arthropod Isoxys minor from the Qingjiang biota)
- Line 67. Correct Isoxys
- Line 70. Should be “among Lagerstätten”
- Lines 78-81. This sentence is not clear. Please reword.
- Lines 89-91. This sentence is not clear. What do you mean by issues? Maybe remove issues and rephrase the sentence.
- Lines 100-101. Do you mean that there are 35 specimens of Pectocaris and an additional one (36 in total)? Or do you mean that there are 34 Pectocaris and 1 Jugatacaris? Please precise. Also please rephrase “and one another” because it is unclear. Also please remove “which are reported here” at the end of the sentence.
- Line 153. Replace “the following” with “many”
- Line 257. Replace “evolving towards adaptations” with “adapting”
- Line 275. Should be “…conditions that were unfavorable for the life of the preservation…”
- Line 304. Please remove “by the way”
- Line 324. Please replace “tender” with “labile”
- Please double-check the typos in the tables. For example, in table one “Nnkown” should be “Unknown” (twice in the table).

In brief, I think that the paper is nice and worthy of publication. Prior to its publication, it needs to be put more in context with existing literature, and the English needs to be checked and corrected in some places.

·

Basic reporting

The article is written in clear, unambiguous, professional English, although I provide recommendations to improve clarity in some sections.
The article has a contextual introduction and background, and the literature is well-referenced and relevant, with minor adjustments.
The structure follows PeerJ standards, although the funding is included in the acknowledgements, rather than in a separate Funding section, as per PeerJ guidelines.
Figures are relevant and their quality is high, well labelled and described, with minor adjustments. 
Raw data supplied (Supplementary Material images).

Experimental design

The research question is well-defined and relevant, and fills a knowledge gap: species of Pectocaris in the Cambrian period and the connection across Chengjiang localities and stratigraphical areas. The methods and observations performed are professional and performed ethically. The methods described are sufficient for replication.

Validity of the findings

All data provided, with conclusions well stated and linked to their research question and supporting results. 

FINAL COMMENTS

The authors Changfei Jin, Hong Chen, Huijuan Mai, Xianguang Hou, Xianfeng Yang and Dayou Zhai provide a professional, well-researched and well-written account of the diversity of Pectocaris in the Xiazhuang section, following previous works of similar quality by the same lead author (Jin et al., 2021 Pap.Pal). 

The study increases our knowledge of the diversity of Cambrian bivalved arthropods and brings upfront an interesting question regarding niche partitioning and functional morphology across Cambrian arthropods, a question that also applies to other Cambrian sites like the Burgess Shale. Furthermore, the authors provide useful faunistic and geological connections across Cambrian sites in southern China. 

For publication, I would recommend:

1. Further strengthening the reasons behind the description of a new species of Pectocaris.
2. Further, explain why the authors consider the carapace to be flipped 180º in the holotype.
3. Revise the systematic section, in particular the order Odaraiida, is valid.
4. Revise some terminology, and ensure is consistent throughout the text: "tail-fan", "carapace ridge", "Canglangpu". 
5. Provide, if possible, a clearer comparison of the endites across Pectocaris.
6. Perform some changes to the text: while the original text can be read clearly, some sentences could use stylistic changes and clarifications.

I believe these changes can be easily performed.

Additional comments

New species guidelines
The new species generally complies with PeerJ and ICZN guidelines, but requires the following: 1) Zoobank registration, 2) LSID number and 3) the following text as per PeerJ guidelines:
"The electronic version of this article in Portable Document Format (PDF) will represent a published work according to the International Commission on Zoological Nomenclature (ICZN), and hence the new names contained in the electronic version are effectively published under that Code from the electronic edition alone. This published work and the nomenclatural acts it contains have been registered in ZooBank, the online registration system for the ICZN. The ZooBank LSIDs (Life Science Identifiers) can be resolved and the associated information viewed through any standard web browser by appending the LSID to the prefix http://zoobank.org/. The LSID for this publication is: [INSERT HERE]. The online version of this work is archived and available from the following digital repositories: PeerJ, PubMed Central SCIE and CLOCKSS." 

The authors carefully justify the presence of a new species and provide helpful arguments in a Differential diagnosis section and a table. Based on the material present at this moment and its state of preservation, a new diagnosis is justified. 

However, given that all Pectocaris species are biogeographically and temporally concurrent, and that some of the diagnostic features could change depending on preservation, I would recommend the authors strengthen their claims with the following:

-A comment on size: the article often mentions "intermediate"-sized. A mention in the description is needed regarding the minimum and maximum size of P. paraspatiosa, as well as for other Pectocaris species. Have the authors considered whether some traits (e.g., the width of the abdomen) could change across development?

-A further description of the differences between carapace shapes between Pectocaris species and Jugatacaris. The abstract mentions the shape of the posterodorsal corner and a transverse carina as diagnostic features, but these traits are not compared with other taxa in the Discussion or in Table 1. An illustration portraying carapace shapes would also help the reader.

-The material for two Pectocaris species was published over a decade ago in black and white, and is difficult to assess the distance between the endites. If the authors could photograph the endites of these two species and create a comparative figure with P. paraspatiosa, this would help strengthen their claims that enditic distance is different. If that is not possible, I recommend providing any morphometric measurement that supports these claims. This is especially important, as endite separation would represent the strongest evidence for niche partitioning.

-A comment regarding the shape of the abdomen, and that the authors have considered a) taphonomic differences, and b) differences potentially attributed to development (e.g., related to size) to ensure that the differences in abdomen width between Pectocaris species are species-diagnostic.


DETAILED COMMENTS

Abstract

L36-by the shaped postero-dorsal corner. Please change to "the shape of the postero-dorsal corner" or explain how is the shape different (does it have a process?...).
L37- transverse carina of the carapace. I'd recommend explaining in the description and discussion about this transverse carina. If this refers to its presence in Jugatacaris (and absence in Pectocaris), I would not include it in the abstract, as it is better to restrict to diagnostic characters to avoid confusion. 
L40-Change to: previous views
L41-Change to: among congeneric species, based on morphological differentiation. (Disparity is the quantification of morphological diversity, so here it is better to mention morphological differentiation, as there is no quantification). 

Introduction
L50-51. There are though, other Pectocaris sp. carapaces from the Balang Formation (PhD thesis by Wen Rongqin, 2019 by the Guizhou University), a body of P. sp. in Haiyan (Yang et al., 2021, Nature Ecology and Evolution). I think it would be good to acknowledge these works. If the authors do not consider these specimens to belong to Pectocaris, please mention why. I think is also good to reference here the presence of Pectocaris in the Hongjingshao by Zeng et al., 2014; which is well-referenced throughout the paper.

L55-56. Please state which is the first publication that classifies them as branchiopod crustaceans (Hou, 1999?). After Hymenocarina. I think is also good to reference sensu Aria & Caron, 2017 (Nature article), as this is the first publication that created a diagnosis of Hymenocarina in our current sense. 

L62-Change to filter-feeders

L67- Change to Isoxys 
L68- I would discourage the use of "periods", given that it may be confusing to some readers, who may read it as a major geological period. And we don't have any concluding evidence of Isoxys present in the Ordovician period. I would recommend writing something like "Some bivalved arthropods, such as Isoxys, are found across the Cambrian period, spanning tens of millions of years", or something along these lines. 
L73-L74: Please refer to my previous comment on the presence of Pectocaris outside the Chengjiang. In any case, the statement "the genus seems to be temporally and spatially restricted" remains true when compared to other motile bivalved arthropods, like Isoxys or Tuzoia (Wen et al., 2019). 
L77: Please clarify whether the Xiaoshiba is the temporal continuation of the Chengjiang. 
L79: Please change to "Lagerstätte, Zeng et al. (2014) mentioned the presence of a specimen..."
L82: I would recommend avoiding the use of "Chengjiang Period", to avoid confusion with a geological period.
L89: Change to: Based on the presence of Pectocaris ... 
L90: Change to: we discuss related issues in

L93: Change to Material and Methods

L109: Change to Systematic Palaeontology

Systematic Palaeontology
L113- Please revise the Order Odaraiida. I don't have access to the original publication by Simonetta and Delle Cave (1975), but subsequent descriptions of odaraiids (e.g., Briggs, 1981) have only mentioned the family Odaraiidae (but not the Order). Similarly, Hou 1999 includes Pectocaris in its own Order, Pectocaridida. 

L144-I would like the authors to further explain why they think the carapace is rotated 180º. In my view, there is no apparent displacement of the carapace, and an upside-down shift, with the dorsal side of the carapace placed ventrally, and still holding the connection to the body is quite unlikely. This is especially true considering that the carapace originates and is connected to the dorsal side of the head- this configuration would mean that the carapace is broken out of the head, but still is connected to the body. Furthermore, its outline can easily be compared with that of Pectocaris inopinata without the rotation. 

L145-To me, one of the strangest sections of the carapace is the ridge. Is the ridge the separation between the two valves, as in Isoxys, is this a lateral ridge as in Jugatacaris, or is it a fold, created by taphonomic compression?

L146-147: Change to: Regarding its preservation, the holotype has the carapace laterally compressed, while the trunk is dorso-ventrally compressed, judging from the shape and width of the telson and the lateral position of the notches which would hold the caudal rami if the specimen was complete. 

L151: and one is preserved in oblique to lateral view, judged from the thin width of the telson and tailfans (please refer to the image in Sup. Material). 

L153: I thank the authors for explaining this (as for comment in line 144). My opinion is the following, and would encourage the authors to address these comments and provide some drawings that explain their thinking processes: 

1) The anterior margin of the carapaces in Pectocaris is usually saggital and the posterior end is broader than the anterior. 
I would recommend explaining what you mean with saggital. I would also recommend not using other species' carapaces as a way to support your diagnostic features. Carapace shapes can change a lot both intra-genus (see Isoxys, Tuzoia), and elements like the shape of the outline are prone to taphonomic deformation (e.g., see Waptia or Canadaspis). The outline of the new species is not complete, and thus, I advise being careful when using it for comparison

2) The dorsal margin is nearly straight while the ventral margin is usually convex.
However, the ventral margin is almost straight in P. inopinata and P. spatiosa. And this is similarly the case in the new species. Furthermore, the ventral outline in the specimen presented, is also partly preserved, based on the images provided here. 

3) The gently curved trunk overlays the anterior margin of the carapace can be better explained as a relative rotation between the trunk and the carapace.
Please indicate which other species of Pectocaris show this rotation, and which images do you refer to. Have the authors considered that maybe on the dorsal side the left valve is not preserved, but the right valve is? This would explain why the trunk overlays one of the valves. 

L163: The diagnosis should go before the preservation section, as it is part of the Systematic Paleontology section. Also, clarify what is the shape of the postero-dorsal edge. 

L165: Change to: towards to. 

L173-174. I would suggest the authors clarify the diagnosis of the rod-like structure. If this is the ridge of the carapace (i.e., the section that unites both valves), then the interpretations provided are wrong, as the dorsal edge of the carapace would not be visible. Given that both left and right carapace valves are visible on the right side of the specimen, I would suggest the ridge is a compression artefact due to the three-dimensionality of the carapace, similar to that described in Jugatacaris (Fu & Zhang, 2011, see "dg"). 

L182-183: There is some discrepancy whether the terminal segment in bivalved arthropods is referred to as "telson" or "terminal segment". In Vannier et al., 2018, the latter is preferred. This is not a major problem, but I would like to point the authors towards this debate. More important, though, is the term "tail-fan". I would suggest replacing this by the term "caudal rami", given that tail-fans often include multiple elements, such as the different tail-fans of radiodonts, and isoxyids, and can be even interpreted as a generic trait encompassing both the caudal rami and the telson. I would suggest, if possible, changing it to caudal rami to be more specific, following the works of Vannier et al., 2018 and Izquierdo-López&Caron, 2019, 2022a).

L182: The sub-triangular sclerites are also found in Fibulacaris (Izquierdo-López&Caron, 2019) and Jugatacaris, and are referred to as "acute lateral extensions" in C97 in the phylogeny by Izquierdo-López&Caron, 2022 (iScience). Given that this is quite a new trait, the authors could keep their nomenclature but would be good to acknowledge the alternative name to easily homologize both terms in the future.

L190: Please indicate in the figure where are the exopodial setae mentioned in this line.

L202: Change to "setae-bearing", the plural of seta.

L218-221: Please change to: "Among the various morphological features described in previous studies (Hou & Sun, 1988; Hou,1999; Hou et al., 2004b; Fu & Zhang, 2011, Jin et al., 2021), the most useful traits for describing the present specimens were: the morphology of the limbs, the abdomen, the telson and the caudal rami." This further clarifies that it is the limbs (not any cephalic appendage) and that the caudal rami are also useful (for example, to differentiate P. inopinata). 

L222: replace telson rami with "caudal rami". 

L223: please change to "limb endites". 

L225: what are "paired grooves"? Do you mean the point of attachment of the caudal rami? If so, would help to add this, as the term "paired groove" has not appeared before in the text.

L228: change to: "represents good evidence of its presence within the present samples". Or along these lines. 

L229-330: Change to: "One specimen is identified as Jugatacaris agilis based on its unidivided caudal rami, a universal characteristic that is absent in Pectocaris (Fig. 4I, J)." 

Ecology

L247-L248; I have commented on this before, but for the reader, it would help to have a comparative figure on the position of the endites across species or some morphometrics that clearly show this differentiation exists across Pectocaris. 

L258-L261: I would suggest removing these lines or adapting it to the comparative description sections, as the ecological significance of some of these characters (carapace, lateral spines, shape of the telson and number of trunk-appendage podomeres) has not been discussed herein. If that is the intention of the authors, I recommend Olesen, 2013 The Crustacean Carapace...for the significance of the carapace. The other traits I listed may not have strong ecological significance.

L263-L265: The authors bring an interesting point about limb specialization and niche differentiation in the Cambrian period, and note that the results reinforce previous conclusions. I would say this is an interesting topic, and I encourage the authors to use further examples to support the view that this is a current topic in the Cambrian. For example, Briggs & Whittington, 1985 describe the modes of life of arthropods at the Burgess Shale. Across bivalved arthropods, many have similar modes of life and geographical distributions (Nereocaris exilis and Nereocaris briggsi, see Legg et al., 2012 and Legg & Caron, 2014; or Fibulacaris and Pakucaris of the Marble Canyon site at the Burgess Shale, see Izquierdo-López&Caron, 2019 and 2021). 

L267: "So-called" appears as an informal term in this type of publication. I recommend using "referred to as" or simply "hymenocarines". 

L268: I recommend not using "Hongjingshao period", rather just "Hongjingshao" or "Hongjingshao Member", to avoid confusion with geological period. 

L269: Briggs, 1994 is outdated in terms of the total diversity of bivalved arthropods. I would recommend using Izquierdo-López&Caron, 2022a (iScience) which has one of the most recent phylogenies focused on bivalved arthropods, including the majority of the species described from the Burgess Shale. 

L273: I would reference other small species from the Chengjiang, like Ercaicunia. 

L275: Change for clarity to: "this may denote stronger water flows that were unfavourable for smaller swimming bivalved arthropods or for their preservation".

Taphonomic implications

L290: Change to: "can also be found"

L304: "by the way" appears too informal. Please change to "Furthermore" or similar. 

L307: Change to "turbulent environments".

L324: Change to "weaker anterior part", rather than tender.

L324-328. Could the authors clarify what type of transportation are they referring to? Based on the previous text, the authors appeared to suggest that the Xiazhuang section is characterized by turbulent waters that would prevent smaller species from either inhabiting or setting post-mortem (L272-276). The authors express that both the Xiazhuang and Yu'anshan specimens are large, thus, supporting this view, and yet, in L326, they express that this event is uncertain. It would help explain how the authors still refer to this as an "uncertain event" despite citing previous support. 

Biostratigraphical connections

L350: Change to "compositions, especially..."

Conclusion
L371: shaped postero-dorsal. In which way is the postero-dorsal edge different?
L380: change to "morphological differentiation", as disparity is a quantification of morphological differences.

Figure captions
If any of the terminology is changed based on the previous comments, please change the figures accordingly. 

L409: what is the difference between Canglangpu, Tsanglangpu and Changlongpu formations, and which nomenclature do you recommend?

References

Table 1

Change Nnknown for Unknown
Align the nomenclature of the table with the nomenclature of the description (before any changes proposed by this reviewer are applied): "shape of caudal segment" should be "telson shape", "telson fluke" should be "tailfan shape".

---

## Round 0.2 · Minor Revisions

Dear authors,

I have now read the revised version of your manuscript about Cambrian arthropod diversity from China, and I saw that it was very improved with respect to the original version. Even so, I have yet some concerns that I hope you can fix quickly. I have prepared an annotated PDF of your manuscript and added some comments there to guide you to make the needed changes. The most important issue to address is the taphonomic section, for which I have made some recommendations in the annotated PDF. As you commented in your rebuttal letter, it is possible that new specimens assignable to Pectocaris can be found outside the Yu’anshan Formation in the future, and so, the comparative morphology of the four described species can be better resolved. Therefore, I would prefer to see less categorical conclusions in this contribution; for instance the differences that you suggest as taxonomically valid relating to a different number of appendages, and the presence/absence of setae, and even the morphology of the telson and the telson processes can be influenced by taphonomy. Also, there can be ontogenetic or even sexual differences that for the time cannot be analyzed from the studied material, but they can be evaluated if other more complete specimens can be analyzed. Moreover, the very common absence of the carapace in the studied specimens can be related to molting behavior and it will be fine to take it into consideration in your discussion. Strong turbulent environments can be right but under extraordinary conditions regarding the presence of appendages and telson preserved in anatomical position. Anywhere, I would be grateful if you consider including my suggestions in the next version of your manuscript and modifying your conclusions to a less categorical interpretations, leaving an open space for complementary diagnoses that may be available for future contributions. With kind regards,
Graciela Piñeiro

---

## Round 0.3 · Minor Revisions

Dear authors,

Great to see that you have a new section on taphonomy in your manuscript. Indeed, I am very gratified to see this last version of your work. I consider that it is almost ready to be in the acceptable form to be published in PeerJ. Before the final acceptance, I have marked just a few modifications that will take a few seconds or a minute for you to fix.

Once the marked changes are done, please resubmit the manuscript for complete the editorial process.

Kind regards,
Graciela Piñeiro

---

## Round 0.4 · accepted · Accept

Dear authors,

Given that the suggested changes were already done, I am pleased to tell you that your manuscript is acceptable for me, and it is ready for publication in PeerJ.

Congratulations!
Best regards,

Graciela Piñeiro